# Using viral diversity to identify HIV-1 variants under HLA-dependent selection in a systematic viral genome-wide screen

Nadia Neuner-Jehle[1,2]*, Marius Zeeb[1,2], Christian W. Thorball[3], Jacques Fellay[3,4,5], Karin J. Metzner[1,2], Paul Frischknecht[1], Kathrin Neumann[1], Christine Leeman[1], Andri Rauch[6], Marcel Stöckle[7], Michael Huber[2], Matthieu Perreau[8], Enos Bernasconi[9], Julia Notter[10], Matthias Hoffmann[11], Karoline Leuzinger[12], Huldrych F. Günthard[1,2], Chloé Pasin[1,2,13©], Roger D. Kouyos[1,2©], the Swiss HIV Cohort Study (SHCS)[¶]

1 Department of Infectious Diseases and Hospital Epidemiology, University Hospital Zurich, Zurich, Switzerland, 2 Institute of Medical Virology, University of Zurich, Zurich, Switzerland, 3 Precision Medicine Unit, Biomedical Data Science Center, Lausanne University Hospital and University of Lausanne, Lausanne, Switzerland, 4 School of Life Sciences, École Polytechnique Fédérale de Lausanne, Lausanne, Switzerland, 5 Swiss Institute of Bioinformatics, Lausanne, Switzerland, 6 Department of Infectious Diseases, Bern University Hospital, University of Bern, Bern, Switzerland, 7 Division of Infectious Diseases and Hospital Epidemiology, University Hospital Basel, University of Basel, Basel, Switzerland, 8 Divisions of Immunology and Allergy, Lausanne University Hospital, Lausanne, Switzerland, 9 Division of Infectious Diseases, Ente Ospedaliero Cantonale, Lugano, University of Geneva and University of Southern Switzerland, Lugano, Switzerland, 10 Division of Infectious Diseases, Infection Prevention and Travel Medicine, Cantonal Hospital St. Gallen, St. Gallen, Switzerland, 11 Division of Infectious Diseases and Hospital Epidemiology, Cantonal Hospital Olten, Olten, Switzerland, 12 Clinical Virology, University Hospital Basel, Basel, Switzerland, 13 Collegium Helveticum, Zurich, Switzerland

© These authors contributed equally to this work.
¶ Membership of Swiss HIV Cohort Study (SHCS) is provided in Supporting Information file S1 Acknowledgments.
* nadia.neuner-jehle@bluewin.ch (NNJ), roger.kouyos@uzh.ch (RDK)

**Data Availability Statement:** The data generated or analyzed during the current study cannot be

## Abstract

The pathogenesis of HIV-1 infection is governed by a highly dynamic, time-dependent interaction between the host and the viral genome. In this study, we developed a novel systematic approach to assess the host-virus interaction, using average pairwise viral diversity as a proxy for time since infection, and applied this method to nearly whole viral genome sequences (n = 4,464), human leukocyte antigen (HLA) genotyping data (n = 1,044), and viral RNA load (VL) measurements during the untreated chronic phase (n = 829) of Swiss HIV Cohort Study participants. Our systematic genome-wide screen revealed for 98 HLA/viral-variant pairs a signature of immune-driven selection in the form of an HLA-dependent effect of infection time on the presence of HIV amino acid variants. Of these pairs, 12 were found to have an effect on VL. Furthermore, 28/58 pairs were validated by time-to-event analyses and 48/92 by computational HLA-epitope predictions. Our diversity-based approach allows a powerful and systematic investigation of the interaction between the virus and cellular immunity, revealing a notable subset of such interaction effects. From an evolutionary perspective, these observations underscore the complexity of HLA-mediated selection pressures on the virus that shape viral evolution and pathogenesis.

shared publicly due to the sensitive nature and privacy concerns (see https://www.shcs.ch/294-open-data-statement-shcs). Investigators with a request for selected data can send a proposal to the Swiss HIV Cohort Study (www.shcs.ch/contact). The provision of data will be evaluated by the Scientific Board of the Swiss HIV Cohort Study and the study team and will be subject to Swiss legal and ethical regulations. The main coding scripts for this study will be made available in a public GitHub repository (under https://github.com/nneune/HLA_APD).

**Funding:** This study has been financed within the framework of the Swiss HIV Cohort Study (SHCS), which is supported by the Swiss National Science Foundation (SNSF; grant #201369), by SHCS project #910, and by the SHCS research foundation. The data are gathered by the Five Swiss University Hospitals, two Cantonal Hospitals, 15 affiliated hospitals, and 36 private physicians (listed in http://www.shcs.ch/180-health-care-providers). This work was further supported by the SNSF (grant numbers 177499 to H.F.G. in the framework of the SHCS; 179571 and 141067 to H.F.G.; and 207957 and 155851 to R.D.K.); the Yvonne-Jacob Foundation (to H.F.G.); the University of Zurich Clinical Research Priority Program for Viral Infectious Disease, the Zurich Primary HIV Infection Cohort Study (to H.F.G.); and an unrestricted research grant from Gilead Sciences (to the SHCS Research Foundation). N.N.-J.'s salary was financed in part by grants from the SNSF (207957) and the SHCS (project #910). C.P. was supported by a fellowship from the Collegium Helveticum. The funders had no role in study design, data collection and analysis, decision to publish, or preparation of the manuscript.

**Competing interests:** K.J.M. has received travel grants and honoraria from Gilead Sciences, Roche Diagnostics, GlaxoSmithKline, Merck Sharp & Dohme, Bristol-Myers Squibb, ViiV, and Abbott; and the University of Zurich has received research grants from Gilead Science, Novartis, Roche, and Merck Sharp & Dohme for studies in which K.J.M. serves as principal investigator, and advisory board honoraria from Gilead Sciences and ViiV. A.R. reports support to his institution for advisory board and/or travel grants from MSD, Gilead Sciences, Pfizer, and Moderna, and an investigator-initiated trial (IIT) grant from Gilead Sciences. All honoraria went to his home institution, not to A.R. personally, and all honoraria were provided outside of the submitted work. M.S. reports advisory board consultations from Gilead, ViiV, MSD, paid to his institution, and travel grants for conferences from Gilead, paid to his institution. E.B.'s institution has

## Author summary

The intricate interplay between viruses and the human immune system is reflected in dynamic associations between the viral and the human genomes. These often take the form of escape dynamics, in which the virus acquires mutations that allow it to evade immune recognition. We developed a novel viral diversity-based method to screen for such interactions across the viral genome systematically and applied it to a unique dataset of HIV-1 sequences and human leukocyte antigen (HLA) variants. We could identify time-dependent interactions between 98 pairs of HLA and viral variants. Among these pairs, 12 were associated with the concentration of viral RNA, longitudinal time-to-event analyses confirmed 28, and 48 were consistent with computational predictions of viral peptide binding to HLA molecules. Our results highlight how the highly dynamic interaction between the viral genome and the immune system shapes viral evolution, and our approach offers new opportunities to systematically study such interactions from real-world cross-sectional data.

## Introduction

The rapid evolution and high diversity of HIV pose challenges to the development of universal vaccines against the virus. Also, HIV disease progression varies significantly among individuals and has been associated with host factors [1–3], and viral evolution [4]. The human genome, specifically the human leukocyte antigen (HLA) genes which encode the major histocompatibility complex (MHC), has emerged as a consistent predictor of HIV disease progression [3,5–9]. HLA class I genes (*A*, *B*, and *C*) have been associated with specific HIV mutations that enable viral immune escape by disrupting antigen processing, epitope-MHC binding, or reducing antigen recognition by T cell receptors [6,9]. These mutations provide the virus a selective advantage in the presence of a matching HLA allele but may be associated with reduced fitness in its absence [9]. Regarding HLA class II, the current evidence is conflicting: studies have computationally predicted CD4+ T-cell-driven escape [10], but this could not be confirmed *in vitro* [11]. Another study showed that linkage disequilibrium of HLA alleles may explain associations with class II [12].

Genome-wide association studies have explored HLA-HIV genome interactions in the context of antiretroviral therapy(ART)-naïve individuals, finding viral mutations associated with HLA alleles [9,12,13], and identifying "protective" HLA class I alleles (e.g. B*57:01) associated with lower viral load and slower disease progression [3,7,9,12–16].

However, as most of these interactions were found as associations between HIV mutations and HLA alleles in cross-sectional studies, their temporal dynamics remain unclear. Previous longitudinal studies described HIV escape mutations occurring within three years after infection. In the presence of a "protective" HLA allele, mutations occurred more rapidly, and in its absence, they reverted quicker upon transmission [16–20]. These studies, however, focused on specific viral epitopes in small populations, therefore lacking the systematic approach of the genome-wide screens. We developed a novel viral diversity-based approach that allows us to investigate the time-dependent effects of HLA-associated HIV adaptation from cross-sectionally sampled viral genomes.

Based on comprehensive data from the Swiss HIV Cohort Study (SHCS) [21], we applied a previously established and validated viral genetic diversity score (average pairwise diversity, APD [22,23]) as a reliable proxy for time since infection (TSI). Determining the TSI can be

received research grants from Gilead and Merck unrelated to this work; E.B.'s institution has also received consultancy fees and travel grants from Gilead, ViiV, Merck, Pfizer, Astra Zeneca, Moderna, Abbvie, and Ely Lilly. J.N. has received travel grants from Gilead. H.F.G. has received grants from the Yvonne-Jacob Foundation, the Clinical Research Priority Program of the University of Zurich, and Gilead Sciences, and, outside of this study, grants for unrestricted research from the Swiss HIV Cohort Study, the Swiss National Science Foundation, the National Institutes of Health, the Bill and Melinda Gates Foundation, Gilead, and ViiV; and personal fees as a consultant for Merck, ViiV Healthcare, and Gilead Sciences and as a member of the Data and Safety Monitoring Board for Merck. H.F.G.'s institution has received educational grants unrelated to this work from Gilead, ViiV, MSD, Abbvie, Pfizer, and Sandoz. C.P. has received fellowships from the Collegium Helveticum. R.D.K. has received grants from Gilead Sciences, the National Institutes of Health, the Swiss National Science Foundation, and the Swiss HIV Cohort Study. N.N.J. has received support from the Swiss National Science Foundation and the Swiss HIV Cohort Study. All other authors report no potential conflicts of interest.

challenging as people may live with HIV for many years before being diagnosed [22,23]. HIV diversity increases throughout the infection, and this diversity can be used to derive the TSI. Of several methods that utilize viral diversity, APD was characterized by a mean absolute error of less than one year and higher sensitivity and specificity than, for example, the fraction of ambiguous nucleotides, FAN, method [22,23]. We used APD to systematically assess the time-dependence of HLA/HIV-variant associations. This has allowed us to gain novel insights into the evolving relationship between the human immune system and the ever-adapting HIV.

## Results

### Study participants

Among all SHCS participants who met our selection criteria (Fig 1A and Table 1), 1,044 had at least one ART-naïve HIV sample sequenced and their HLA alleles genotyped (analysis Ib), and 829 had VL measurements within 180 days around the sample date (analysis II). For the longitudinal analysis III, 130 participants had 340 sequences available. Most participants were male (86.2–90%), had "White" self-reported ethnicity (93.1–94.4%), and were of European or Northern American origin (90.0–93.8%). The time between the first HIV diagnosis and ART start varied between analysis Ib (median [Interquartile range (IQR)]: 2.15 [0.31, 4.76] years) and analysis II (median [IQR]: 2.93 [1.41, 5.65] years).

### HIV amino acid variants associated with HLA alleles

Among the 1,528 SHCS participants included in statistical analysis Ia, we identified 1,146,205 combinations of HLA alleles and HIV amino acid variants (S1 Fig and S1 Table). We employed a systematic pre-screening to narrow the selection down to HLA/HIV-variant pairs of interest, resulting in the following: First, we identified pairs that yielded a statistical power of at least 80% for detecting an HLA/HIV-variant association with an odds ratio (OR) of three (n = 208,224). Next, we selected those pairs that were associated in a Fisher's exact test (FDR<0.2) (n = 532). Among these 532 associations between HLA alleles and HIV variants, 99 were no longer statistically significant (FDR-corrected p>0.05) when tested in multivariable logistic regression models adjusted for APD and the first ten human genome-based Principal Components (PCs) (S1 Fig and S1 Table). Ancestral and population structure biases likely accounted for these associations. Of the 433 remaining significant pairs, 329 showed a positive association, wherein the viral variant was more prevalent when the HLA allele was present (OR median [IQR]: 2.86 [2.19, 3.83], as shown in S2 Fig).

Most of the HIV variants from these 433 pairs were found in the proteins Pol (n = 131), Nef (n = 89), and Gag (n = 77). Among the HIV proteins considered, Vpr had the highest relative frequency of significant HLA/HIV-variant pairs (with 37 pairs/58,404 possible combinations), followed by Nef (89/165,478), Pol (131/365,496), and Gag (77/233,930). The protein Env had the lowest relative frequency (41/930'696) (S1 Table). Most variants (n = 365) were associated with class I HLA alleles (72 with HLA-A, 174 with HLA-B, and 119 with HLA-C), and the remaining 68 variants with class II HLA alleles (5 with HLA-DPA1, 9 with HLA-DPB1, 14 with HLA-DQA1, 25 with HLA-DQB1 and 15 with HLA-DRB1) (S2 Fig). The strongest HLA/HIV-variant association was found for HLA-B*57:01 and Gag242N (OR = 102.67, 95% CI [47.37, 247.06], p<0.001). Five other HLA alleles (A*01:01, C*06:02, DRB1*07:01, DQA1*02:01, and DQB1*03:03) were associated with Gag242N as well (OR median [IQR]: 3.102 [2.11, 5.37]). For Pol, the strongest association was found between HLA-B*15:01 and 149L (PR 93L) (OR = 7.88, 95% CI [4.71, 13.80], p<0.001), and for Nef the strongest association was HLA-A*11:01 with 92R (OR = 16.17, 95% CI [9.53, 27.69], p<0.001).

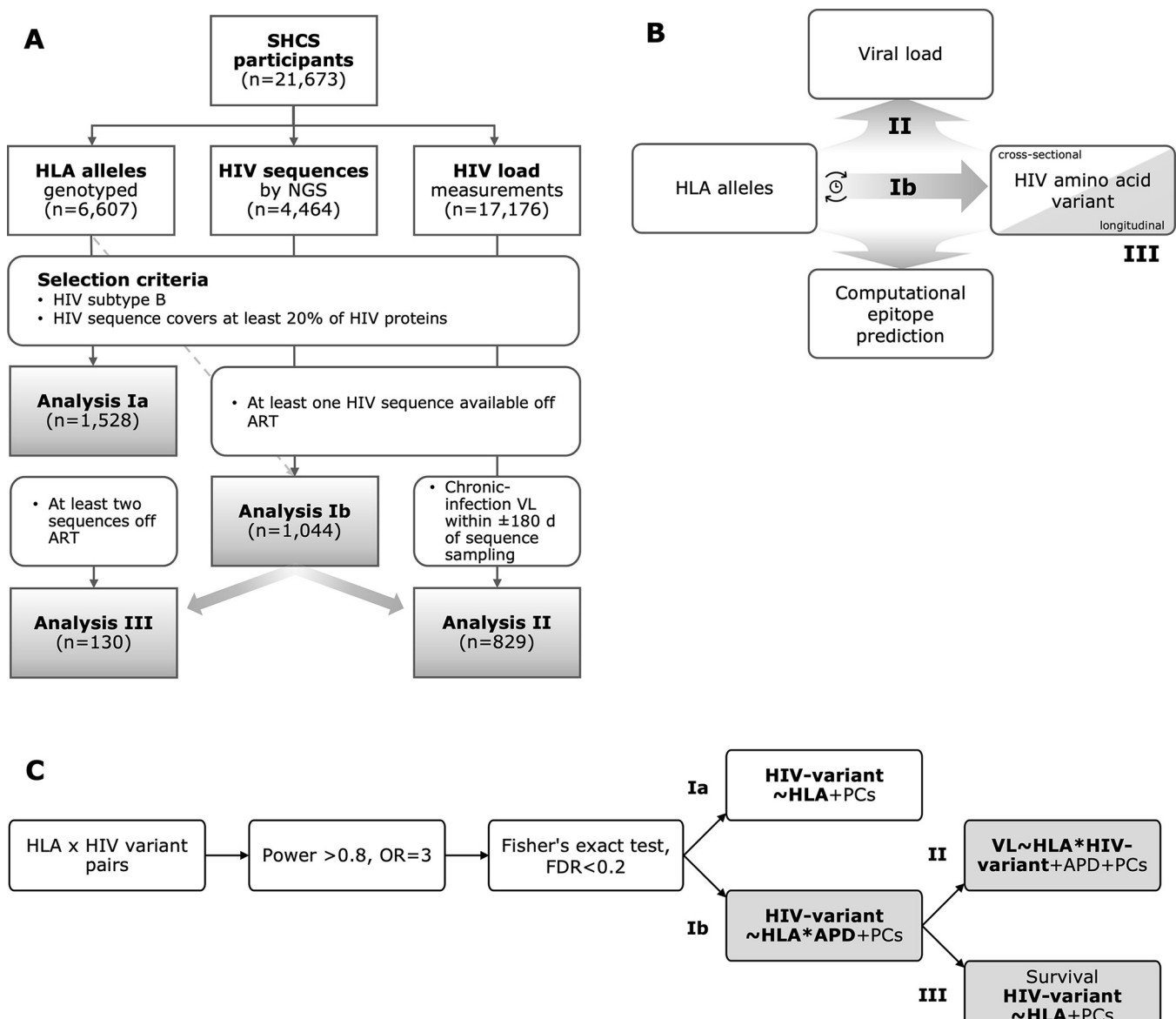

**Fig 1. Statistical analysis plan with participant selection.** A) Measurement data selection of Swiss HIV Cohort Study (SHCS) participants with human leukocyte antigen (HLA) alleles and next-generation sequencing (NGS) data for analysis Ia & Ib, HIV load (VL) data for analysis II, and multiple sequences for analysis III. B) HIV amino acid variants are tested in logistic regressions for their association with HLA alleles and their interaction effect between HLA alleles and average pairwise diversity (APD) (analysis Ib). Pairs with significant interaction in analysis Ib were further tested for their effect on VL (analysis II), retested in a longitudinal setting (analysis III), and assessed by computational HLA-epitope binding predictions. C) Pairs were selected if power>0.8 and Fisher's exact test p-value<0.2 (FDR-corrected). Statistical analyses performed: [Ia] Presence of HIV variants as function of HLA alleles. [Ib] Presence of HIV variants as function of HLA alleles, APD, and interaction between HLA and APD. [II] VL levels as function of HLA/HIV-variant pairs (identified as significantly associated in [Ib]); and [III] longitudinal survival analysis of HLA/HIV-variant pairs identified in [Ib]. Only samples from ART-naïve participants were used for the grey-shaded analyses ([Ib], [II] and [III]).

## Evaluating HLA-dependent selection for HIV variants reveals 98 time-dependent pairs

To assess the time dependence of HLA selection on HIV variants, we evaluated the interaction between HLA and APD on the odds of observing the viral variant. Among the 1,044 SHCS

**Table 1. Characteristics of SHCS participants included.** The number of participants used for cross-sectional (I and II) and longitudinal analysis (III) and their characteristics are shown for each analysis, restricted to subtype B.

| | Analysis Ia | Analysis Ib | Analysis II | Analysis III |
|---|---|---|---|---|
| **Sample type** | HLA, HIV seq | HLA, HIV seq | VL, HLA, HIV seq | HLA, HIV seq |
| **Number of SHCS participants** | 1528 | 1044 | 829 | 130 |
| **ART-naïve:** n (%) | 1044 (68.3) | 1044 (100.0) | 829 (100.0) | 130 (100.0) |
| **Sex:** Male (%) | 1317 (86.2) | 914 (87.5) | 719 (86.7) | 117 (90.0) |
| **Baseline Age*:** median [IQR] | 37 [31, 44] | 36.5 [30, 43] | 36 [30, 42] | 36 [30, 41] |
| **Region:** n (%) | | | | |
| Europe & North America | 1424 (93.2) | 979 (93.8) | 772 (93.1) | 117 (90.0) |
| Africa | 23 (1.5) | 17 (1.6) | 15 (1.8) | 2 (1.5) |
| Middle & South America | 52 (3.4) | 26 (2.5) | 24 (2.9) | 6 (4.6) |
| Asia | 28 (1.8) | 21 (2.0) | 17 (2.1) | 4 (3.1) |
| other | 1 (0.1) | 1 (0.1) | 1 (0.1) | 1 (0.8) |
| **Ethnicity:** n (%) | | | | |
| White | 1443 (94.4) | 985 (94.3) | 778 (93.8) | 121 (93.1) |
| Black | 29 (1.9) | 19 (1.8) | 18 (2.2) | 2 (1.5) |
| Hispano-American | 27 (1.8) | 19 (1.8) | 16 (1.9) | 5 (3.8) |
| Asian | 27 (1.8) | 20 (1.9) | 17 (2.1) | 2 (1.5) |
| other | 2 (0.1) | 1 (0.1) | 0 (0.0) | 0 (0.0) |
| **HIV $\log_{10}$ RNA:** median [IQR] | - | - | 4.48 [4.04, 4.95] | - |
| **CD4$^+$ T cell count:** median [IQR] | - | - | 401.0 [309.5, 519.0] | - |
| **proviral DNA:** n (%) | 484 (31.7) | 61 (5.8) | 36 (4.3) | 1 (0.8) |
| **APD$_z$ score:** median [IQR] | 0.63 [0.26, 1.23] | 0.60 [0.29, 1.14] | 0.64 [0.35, 1.19] | 0.47 [0.30, 0.82] |
| **Years until ART:** median [IQR] | 1.65 [0.10, 4.58] | 2.15 [0.31, 4.76] | .93 [1.41, 5.65] | 2.85 [2.14, 4.67] |

*Age at the SHCS registration date. Abbreviations: HLA = Human Leukocyte Antigen, HIV seq = HIV sequence, VL = viral load, IQR = Interquartile range, APDz = standardized Average Pairwise Diversity, ART = Antiretroviral therapy, SHCS = Swiss HIV Cohort Study.

participants included in statistical analysis Ib, we found 98 significant HLA/HIV-variant pairs among all 433 pairs with significant associations in Fisher's exact tests (Fig 2A and S2 Table). Most of these pairs (84/98) had a positive interaction effect between HLA allele and APD (OR IQR [1.71,2.28]), i.e., a decreasing chance of HIV variant occurrence over time was observed in the absence of the HLA allele and an increasing chance was observed in its presence, indicating an HLA-dependent selection pressure on the viral variant (Fig 2D). The amino acid variant Pol432R is an example of this larger increase over time in the presence of HLA-A*03:01. For this pair, the HLA-APD interaction term was positive (OR 3.62, 95% CI [2.09, 6.94], $p<0.001$) (Fig 2B). In the absence of HLA-A*03:01, Pol432R was less likely to occur with increasing APD (OR 0.69, 95% CI [0.58, 0.81], $p<0.001$), but in the presence of the allele, this trend was reversed (OR 2.50, 95% CI [1.41, 4.45], $p = 0.002$) (Fig 2C).

Notably, of the remaining 14/98 pairs with significant HLA-APD interactions, four remained stable in absence of the HLA allele, and ten pairs even had negative interaction terms (Fig 2D and S2 Table). An example for this negative HLA/HIV-variant association and APD-HLA interaction effect is the variant Rev57E, as it was more evident over time in the absence of HLA-B*40:01 (Fig 2B and 2C). For 335 pairs, a significant HLA/HIV-variant association but no significant interaction with APD was estimated. For example, the associated pair Gag242N:B*57:01 (Fig 2B) showed no interaction with APD, suggesting a robust and early selection of Gag242N in HLA-B*57:01-carriers.

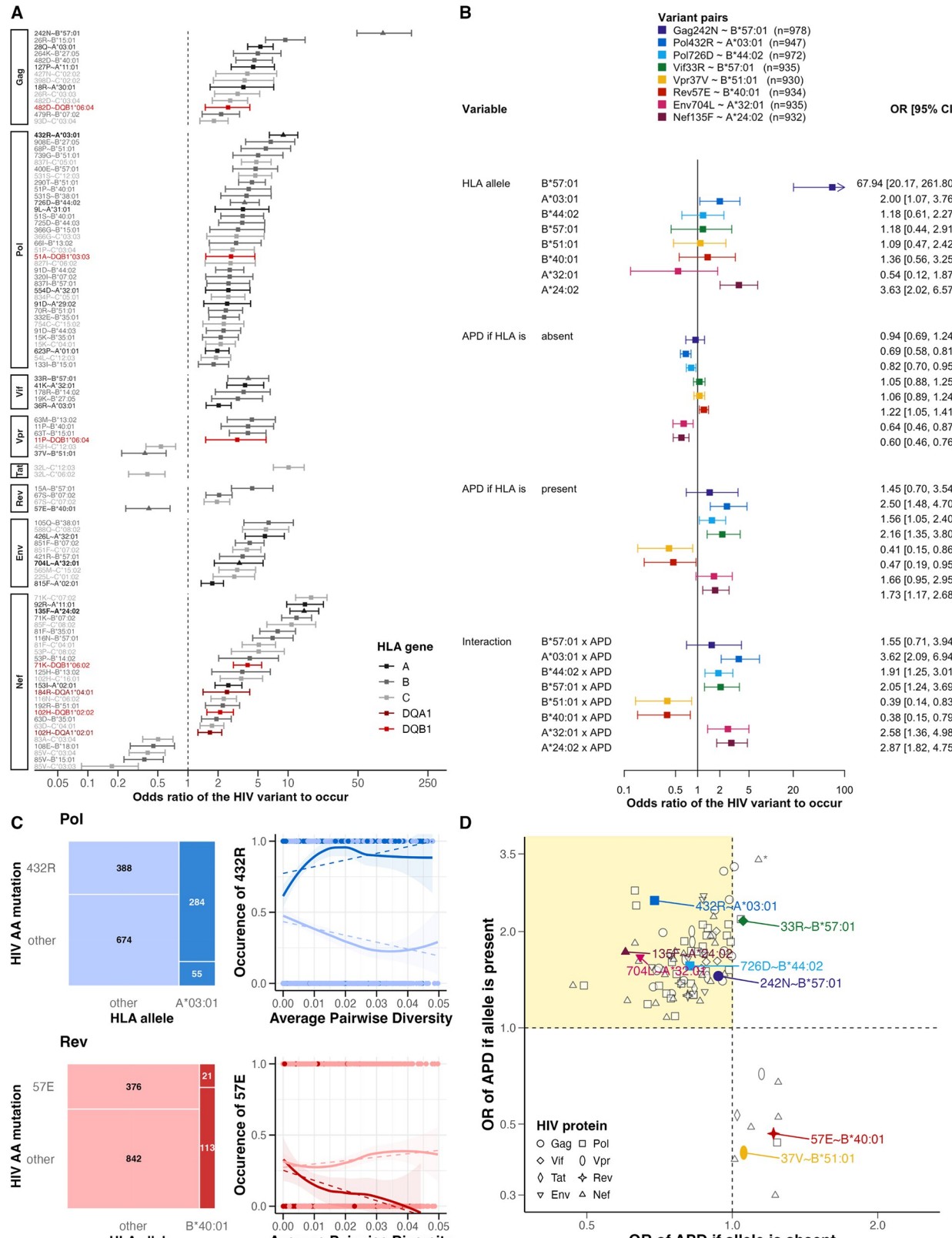

**Fig 2. Associations between HIV amino acid variants, HLA alleles, and average pairwise diversity interactions.** A) Shown are Gag242N: B*57:01 and 98 HLA/HIV-variant pairs with significant average pairwise diversity (APD)-HLA interactions (analysis Iβ). Odds ratios (OR), sorted

by effect size and HIV protein, are derived from Fisher's exact tests, and colored by HLA genes (class I gray, class II red). Pairs of interest are highlighted (bold label, triangle symbol). B) Multivariable logistic regressions of eight HLA/HIV-variant pairs of interest are shown. The effect of HLA allele, APD, and their interaction on the occurrence of the viral variant is presented as odds ratios with 95% confidence intervals (OR [CI]). Sample sizes are indicated (n). C) Distribution of the pairs Pol432R:A*03:01 (blue) and Rev57E:B*40:01 (red) within the study population and the likelihood of observing the variant in the presence (dark color) or absence (light color) of the HLA allele over APD is shown. Labels indicate the sample sizes for each group. Dashed lines represent the fitted values of linear regressions, while plain lines show data with standard error. D) Effect of APD on presence of viral variant (estimated OR from analysis Iβ) in presence/absence of HLA allele. Coloration according to panel B. Positive interaction effects in yellow quadrant (= expected HLA escape mutation). *All estimates exceeding the third quartile+1.5 were adjusted to this threshold. Standardized APD used in panel B and D.

## Twelve HLA/HIV-variant pairs affected HIV viral load

In linear regression models, HLA-B*57:01 and B*57:03 were highly associated with lower VL ($p<0.001$, S3 Fig) and showed significant interaction effects on VL with viral variants of the proteins Gag, Nef, Pol, and Vif. Assessing those effects among all 98 pairs with significant interaction effects between APD and HLA on the viral variant in analysis II, we found 12 HLA/HIV-variant pairs in total with a significant interaction effect on VL (IQR regression coefficient (β) [0.43,0.61]) (Figs 3D and 3E and S2 Table). All but one of the twelve pairs showed a positive interaction effect on VL, i.e. the VL was higher in HLA-carriers with the mutation and lower in HLA-carriers without the mutation. For instance, those participants with the HLA-A*03:01 allele had a significantly lower VL than those without it (β = -0.45, 95% CI [-0.81,-0.10], $p = 0.013$). However, when Pol432R was acquired, VL increased again in HLA-A*03:01-carriers (β = 0.60, 95% CI [0.22, 0.98], $p = 0.002$) (Fig 3A and 3B). Interestingly, the Gag242N:B*57:01 pair exhibited no HLA-APD interaction effect, yet demonstrated increased VL levels upon mutation, following the trend of the other eleven pairs with positive interactions (Fig 3A and 3E). The pair Rev57E:B*40:01, with a negative interaction effect on VL, showed increased VL levels in the presence of HLA-B*40:01 (β = 0.31, 95% CI [0.10, 0.52], $p = 0.003$), but upon mutation Rev57E, the VL decreased again (β = -0.53, 95% CI [-1.00, -0.06], $p = 0.027$) (Fig 3A and 3C). Overall, we saw that the interaction effects of HLA and APD on the HIV variant corresponded with the HLA/HIV-variant interaction effects on VL ($\rho_{spearman} = 0.94$, $p<0.001$ for significant HIV-HLA interactions; S5A Fig).

## Time-to-event analyses confirm previous associations of HLA with viral APD

Using longitudinally sampled HIV sequences, we evaluated the impact of HLA alleles on the hazards of HIV variant occurrence in analysis III. We excluded 20 pairs of the 98 previously identified HLA/HIV-variant pairs with a significant HLA-APD interaction (Fig 2B and 3D), that had less than two events (i.e. incident mutation) and 20 pairs that did not have at least one event per exposure group (presence/absence of HLA allele). Cox-proportional hazards-regression models revealed significant differences in the hazard of acquiring the viral variant over time in the presence or absence of the HLA allele for 28 of the remaining 58 pairs (hazard ratio [HR] IQR [7.06, 16.99]). The median time to mutation was shorter in the presence of HLA (median [IQR]: 1.36 [1.04, 1.65] years) compared to the participants without the HLA allele (median [IQR]: 1.44 [1.29, 1.48] years). Overall, the correlation between the APD-HLA interaction ORs and the HR of the longitudinal analysis was moderate ($\rho_{spearman} = 0.48$, $p<0.001$; S5B Fig), and the directionality of the significant HRs was consistent with the directionality of the ORs (Fig 4A).

Two examples illustrate the range of patterns (Fig 4B and 4C): Pol-432R developed quickly with an incidence rate of 32.3/100 person-years in participants with HLA-A*03:01, whereas the mutation occurred more slowly in participants with another HLA-A allele (4.75/100

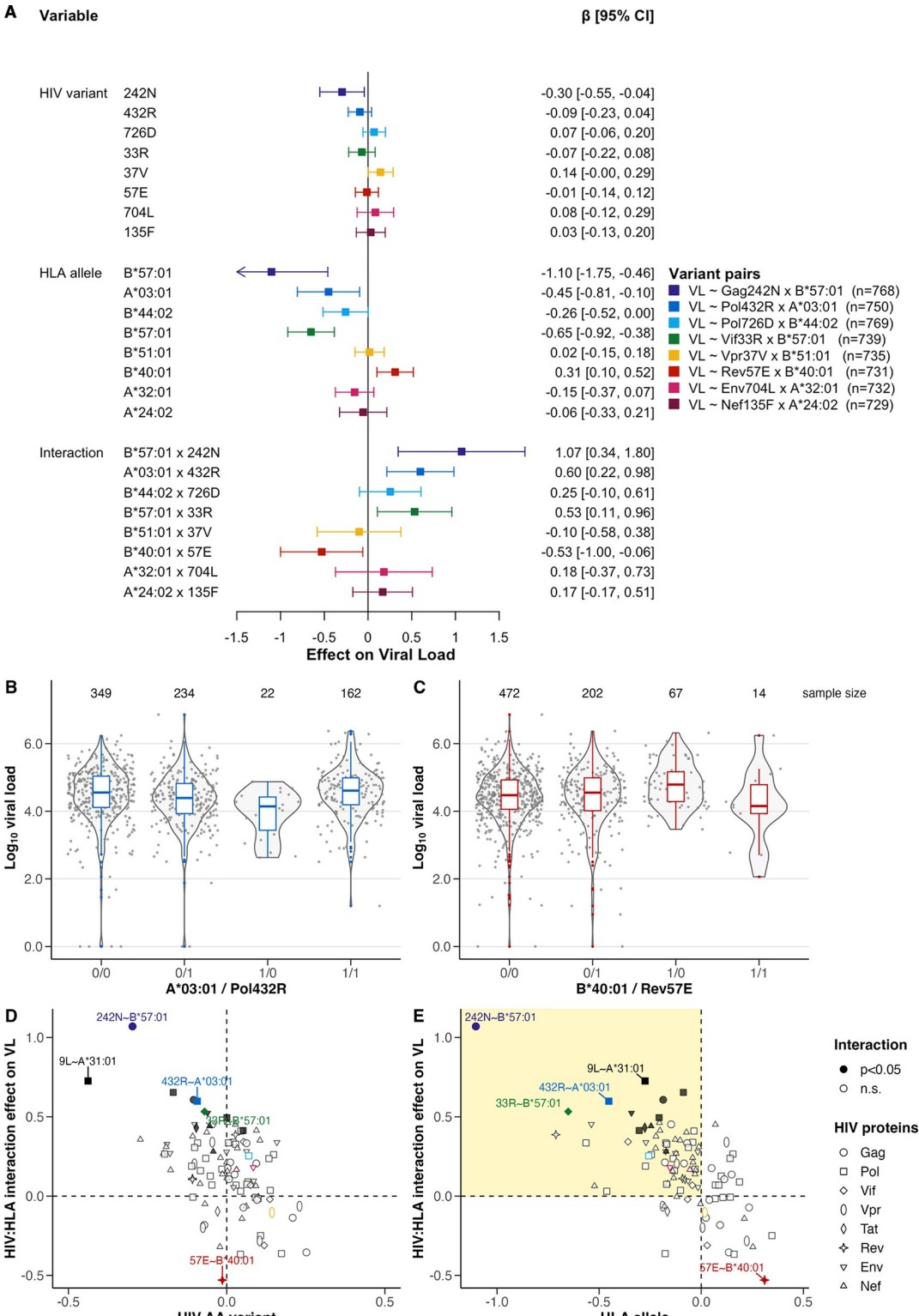

**Fig 3. Viral load association with human leukocyte antigen and HIV variant interaction.** A) Multivariate linear regressions of eight HLA/HIV-variant pairs of interest (same as in Fig 1B), depicted as their association and interaction effect on viral load (VL). B, C) VL distribution over four groups for pair Pol432R:A*03:01 (B) and Rev57E:B*40:01 (C). The absence of human leukocyte antigen (HLA) or HIV variant is coded as '0' and the presence as '1'. Each participant is represented by one point (mean log10 VL), and violin plots show distribution, and boxplots show median and IQR of the distribution. Coloration

according to panel A. D, E) Multiple linear regression estimates of HIV amino acid (AA) variant on VL (D) or HLA allele on VL (E) and HLA/HIV-variant interaction on VL plotted for Gag242N:B*57:01 and all 98 HLA/HIV-variant pairs with significant average pairwise diversity (APD)-HLA interactions (analysis Iβ). The 12 HLA/HIV-variant pairs with a significant interaction effect on viral load ($p<0.05$) are represented by whole symbols and all non-significant (n.s.) by empty symbols. Shapes indicate HIV protein. Coloration according to panel A. Upper-left section in E (yellow) indicates expected HLA escape mutations.

person-years; HR 6.75, 95% CI [1.74, 26.19], $p = 0.006$) (Fig 4B). On the other hand, Rev57E arose more quickly in the absence of HLA-B*40:01 and not at all in the presence of the allele (Fig 4C).

## Computational epitope predictions confirm 48 MHC escape mutations

We used computational MHC class I binding prediction, to assess whether the identified HLA-allele/viral-variants pairs could be explained mechanistically. Among the 92 pairs tested (excluding MHC class II, n = 6), 66 pairs (98 epitopes) were predicted to exhibit at least "weak" binding of at least one non-mutated or mutated 9-mer epitope to the corresponding HLA allele (see S2 Table for more information on epitopes). For these, the impact of the mutation on predicted binding was quantified as rank ratios between mutation and consensus (non-mutated) sequence (Fig 5A). Compared to the HLA-APD interaction effects inferred in the analysis I, 48/66 pairs exhibited epitopes with effects in the same direction as the interaction effects, i.e., weaker binding upon mutation for escape mutations (42/48) and vice versa for mutations with a negative HLA-APD interaction effect (6/48), and 18 pairs had epitopes with only divergent effects, i.e., stronger binding upon mutation despite escape mutation characteristic (Figs 5A and S5C). Overall, there was a weak correlation between the APD-HLA interaction effects and rank ratios ($\rho_{spearman} = 0.11$, $p = 0.38$; S5C Fig). For 56/98 epitopes (41/66 pairs), the mutation did not change the binding category (weak/strong binding) corresponding to the binding ranks (S4 Fig). For 24 (22 pairs) the binding category was predicted weaker upon mutation, e.g., Pol432R:A*03:01 position 9 (still strong binder; Fig 5B), and for 18 epitopes (17 pairs) the binding category was predicted stronger upon mutation, e.g., Rev57E: B*40:01 position 2 (Fig 5C).

## Sensitivity analyses

As statistical analysis Ib was restricted to HIV-1 subtype B and included proviral sequences, we tested in a sensitivity analysis the impact of subtypes and sequence origin on the HLA-APD interaction for the 98 identified pairs. When excluding the 61 proviral DNA sequences, we observed almost identical results ($\rho_{spearman} = 0.96$, $p<0.001$; S5D Fig). Whilst inferring HLA-APD interaction effects for 320 HIV-1 non-B subtype sequences (see S3 Table), we observed a moderate correlation with the effects for subtype B ($\rho = 0.34$, $p = 0.005$; S5E Fig) and saw that five pairs retained significance, reflecting potential subtype differences, and limited statistical power for non-B subtypes. When the sequences of all subtypes (B and non-B) were pooled, a strong correlation was observed with the effects for subtype B ($\rho = 0.88$, $p<0.001$; S5F Fig). In the pooled analysis, the majority of associations (80 of 98) from the subtype B-only analysis (Ib) were recovered. All 80 pairs exhibited significant HLA-APD interaction effects even after adjusting for the first ten viral PCs, thereby accounting for viral population structure. Lastly, HLA alleles associated with the same viral variant were examined in multivariable logistic regressions to test for the effect of linkage disequilibrium among HLA alleles. In these analyses, none of the class II HLA alleles retained their significant interaction with APD (S6 Fig).

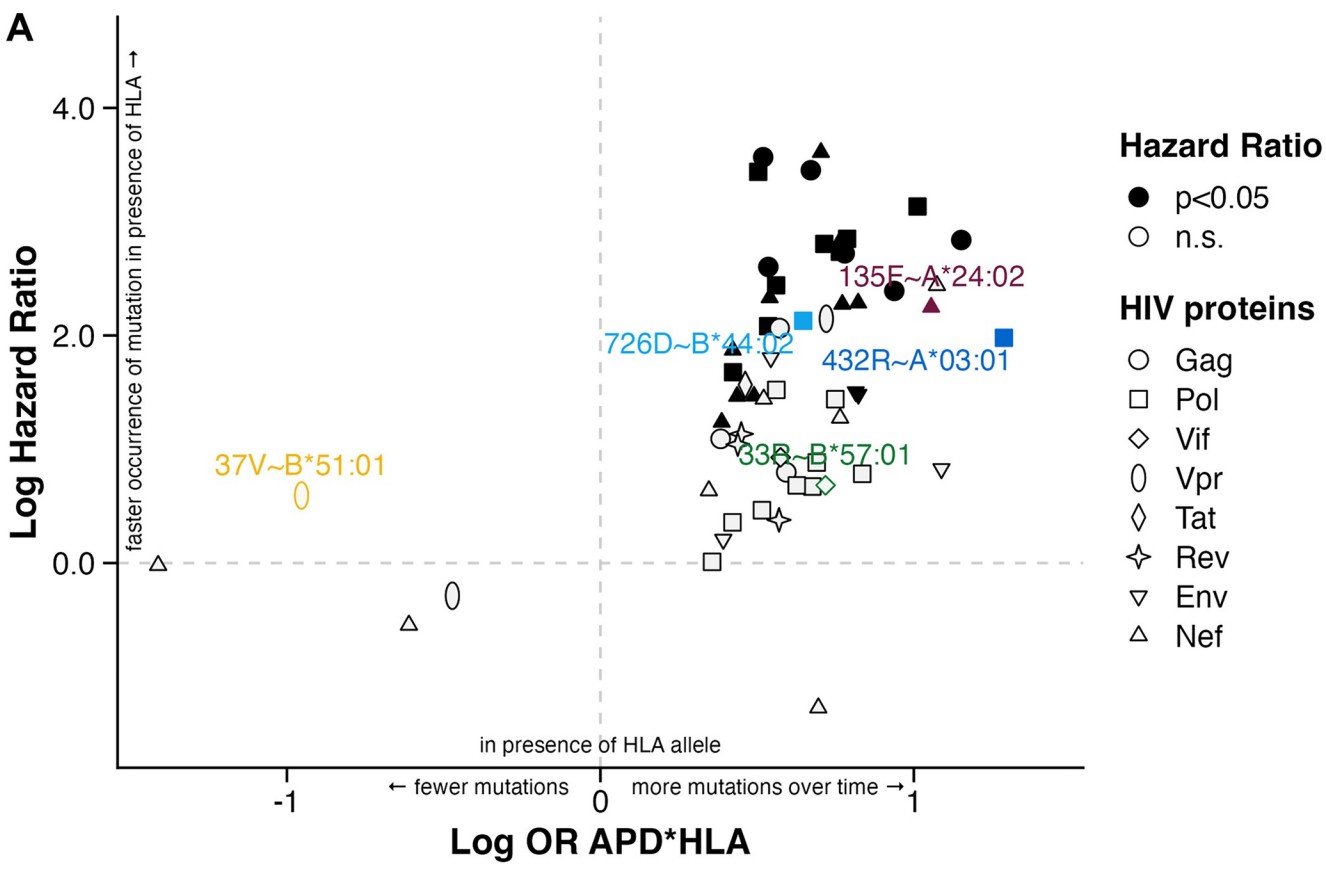

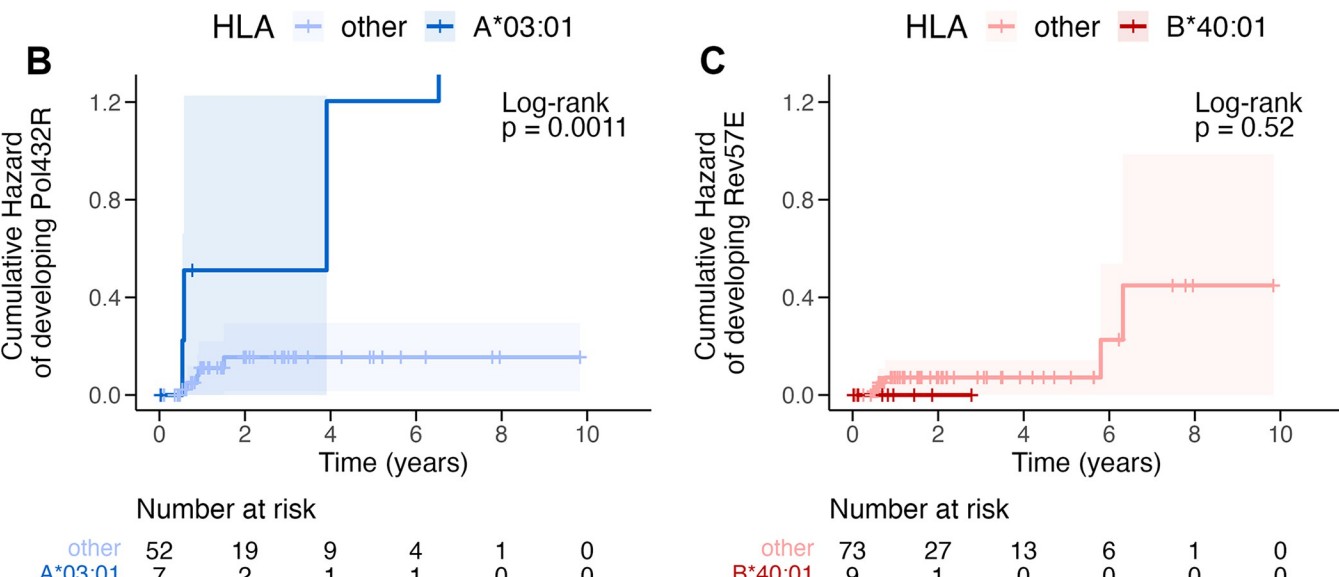

**Fig 4. Cumulative hazard of developing an amino acid variant over time concerning the presence of HLA alleles.** A) Log odds ratios (OR) of interaction effects between APD and HLA on the viral variant versus the Log Hazard Ratio of the respective pair. Significant HLA/HIV-variant interactions (p<0.05) are represented by whole and non-significant (n.s.) by empty symbols. HLA/HIV-variant pairs of interest (same as in Fig 1B) are highlighted. B, C) Cumulative Hazards of acquiring HIV variant Pol432R (B) or Rev57E (C) in presence (dark blue (B) or dark red (C)) of respective HLA allele versus its absence (light blue (B) or light red (C)), are indicated by the respective lines, confidence intervals are illustrated by the shaded area. Censored events are displayed as vertical lines. P-values are shown as log-rank. "Number at risk" indicates the number of participants with outstanding events after the specified time in years.

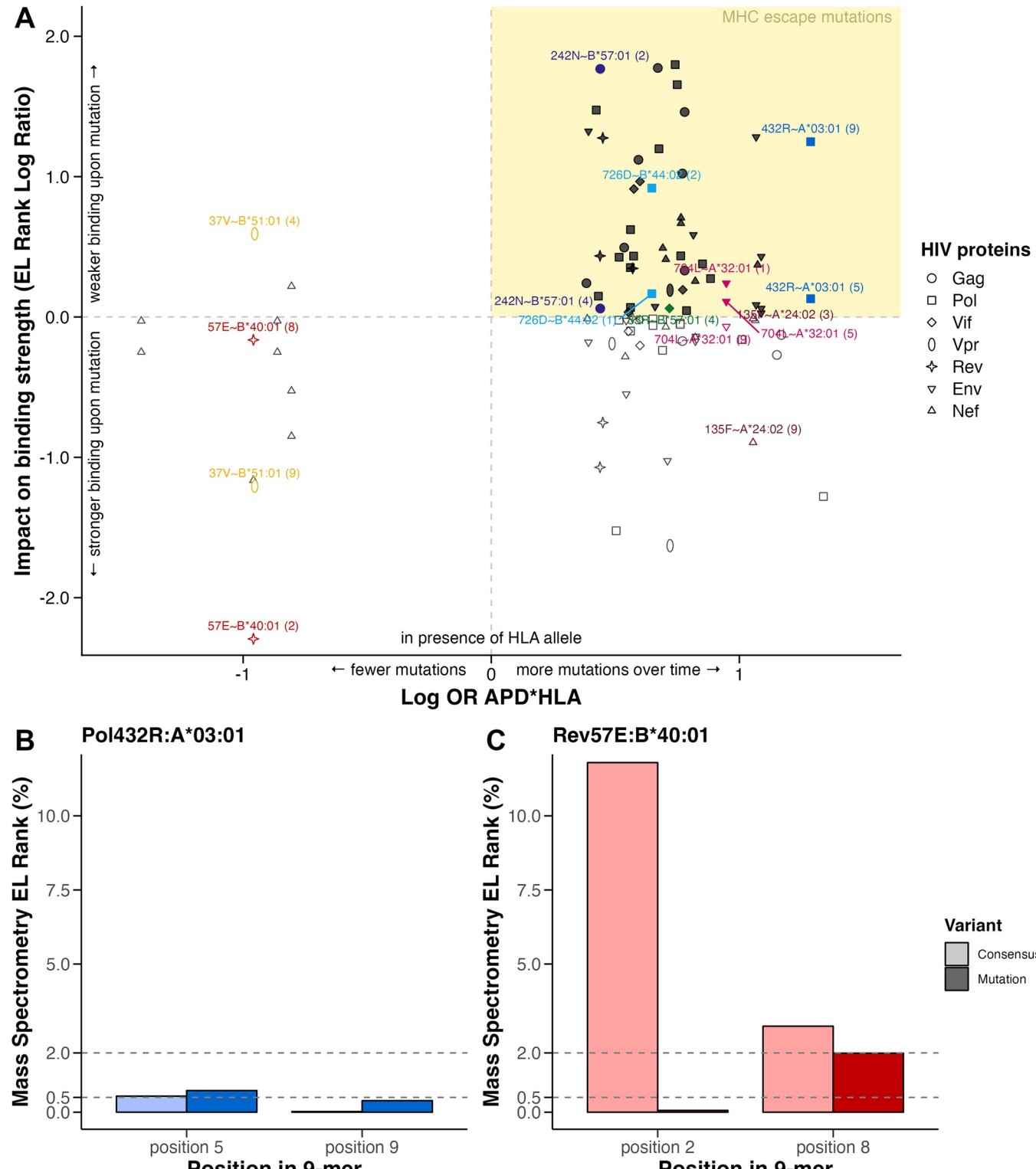

**Fig 5. HLA-HIV peptide binding prediction based on mass spectrometry eluted ligands (EL) rank (%).** A) Log odds ratio (OR) of the Average Pairwise Diversity (APD)-HLA interaction versus the Log Ratio of EL Rank of mutation/EL Rank of consensus. Upper-right section (yellow) indicates expected major histocompatibility complex (MHC) escape mutations. Labels describe HLA/HIV-variant pairs of interest (same as in Fig 1B) and the epitope position of mutation (ranging from 1 to 9). B, C) Peptides are 9-mer HIV sequences, including the mutation (dark color; Pol432R/ RT277R (B) or Rev57E (C)) or consensus (light color; Pol432K/ RT277K (B) or Rev57G (C)) at different positions (ranging from 1 to 9). The rank was categorized into strong binder (≤0.5%),

weak binder ($\leq$2.0%), and non-binder ($>$2.0%). Non-binding positions are omitted here. All binding prediction computations are derived from NetMHCpan-4.1.

## Discussion

Viral and human genetic variants interact in an intricate, highly dynamic way, and viral variants may increase or decrease in frequency during an infection depending on the host genome [3, 4]. Combining large-scale genome-wide screens [6, 7, 12–14] with a study of these adaptive dynamics [16, 17] is challenging as it usually requires longitudinally sampled treatment-naive sequences. This is often prohibitive for large patient populations, especially in the era of immediate treatment initiation. To overcome these limitations, we developed a viral diversity-based approach based on which we quantified the HLA-dependent dynamics of viral variants in over a thousand people with HIV.

This approach stands out by harnessing the potential of cross-sectional data and hence bridging between the within-host and epidemiological perspectives. We identified a significant HLA-dependent effect of infection time (approximated by APD) on the frequency of 98 individual HLA/HIV-variant pairs as a signature for HLA-dependent selection. Of these, 84 exhibited a pattern to be expected of HLA-escape mutations, i.e., a decreasing frequency over infection time in the absence of the HLA allele and an increasing frequency in its presence. We validated and contextualized these results by assessing the dynamics of viral variants in longitudinally sampled SHCS participants and by testing to what extent the identified viral variants had an HLA-dependent impact on VL and disrupted binding to the corresponding HLA alleles.

The time-to-event analysis revealed the robustness of our findings and underscored the limitations of longitudinal data regarding statistical power and scarcity of observed incident viral mutations. About one-third (28/98) of escape mutations found by the APD-based approach could be confirmed in time-to-event analyses and the corresponding effect sizes of the diversity-based and longitudinal analyses were significantly correlated ($\rho_{spearman}$ = 0.48, $p<$0.001). Almost half (40/98) of the pairs could not be assessed in the analysis because they had too small numbers of incident mutations. Previous studies circumvented the limited power of such longitudinal data by limiting their analyses to specific viral epitopes and/or HLA alleles, potentially missing associations revealed by our systematic screening of the cross-sectional data [16,17,24].

Twelve HLA/HIV-variant pairs had a significant HLA-dependent effect on VL and the magnitude of their effect on VL was strongly correlated with the magnitude of the HLA-dependent selection inferred in the diversity-based screen ($\rho$ = 0.94, $p<$0.001). Most (11/12) of these interaction effects resulted in increased VL, consistent with previous findings [12,13] and their classification as HLA-escape mutations. The increase in VL can be explained by the fact that HLA alleles such as B*57:01, B*57:03, or C*12:03 lose their "protective" effect (lowering VL) when an escape mutation like Gag242N occurs [12,13,15]. This is especially relevant when the escape mutation has low fitness costs or reversion rates, as we saw for example with Vif33R and B*57:01 (Figs 2D and 3D). At the population level, this could enhance infectivity and pathogenicity [25, 26], similar to what was described in a Japanese study with homogeneous A*24:02 distribution selecting for the escape mutation Nef135F [27].

Of the twelve pairs that significantly interacted with VL, only the two pairs Vif33R:B*57:01 and Pol432R:A*03:01 were previously described as significantly lowering VL [12,13]. For all others, this interaction effect on VL was either not significant (Gag-264K, Pol54L, Pol68P, Pol91D, and Tat32L) [13] or never identified before (Pol9L, Rev57E, and Nef102H).

Conversely, for the newly identified pair Rev57E:B*40:01, we observed an HLA-dependent decrease in VL upon mutation. Rev57E accumulated over time in the absence of B*40:01 but disappeared in its presence, and binding predictions indicated that Rev57E induced a shift from non-binding to strong binding to B*40:01. This suggests that Rev57E increases viral fitness in the absence of B*40:01 but exposes a B*40:01-dependent epitope causing a fitness cost in the presence of B*40:01. These findings offer insights into potential epitopes with high mutational barriers and/or deleterious mutations, which may warrant attention in vaccine design efforts.

The identification of "only" 433 HLA/HIV-variant pairs out of over two million possible combinations may appear modest at first glance. However, it is important to note that it is to be expected that only a minority of positions in the viral genome interact with HLA, due to the selectivity of HLA binding, immuno-dominance, restriction by the proteasome, and T cell receptors, as well as constraints to viral evolution [6, 9]. Moreover, our analysis revealed that even fewer pairs exhibited a time-dependent relationship (98/433), with an even smaller subset demonstrating a significant effect on VL (12/98). It is possible that some of the 433 pairs may be due to spurious associations resulting from population structure and founding effects [28]. However, such confounding by population structure cannot explain the HLA-APD interactions observed in 98 pairs (analysis Ib) and the effect of HLA in within-patient survival analyses (analysis III), as these reflect HLA-dependent impacts of time since infection on mutation frequency. This robustness is underlined by the fact that most of these pairs are robust to adjustment for viral population structure (S5F Fig), and hence it represents an additional advantage of the viral diversity-based method established here. In fact, this confounding may be one potential additional reason behind the discrepancy between the number of pairs (n = 335) lost from analysis Ia to Ib.

These findings underscore the complexity of the host-virus interaction landscape, and HIV's ability to adapt and evolve in response to immune pressure. This is particularly evident in the observation that for the majority of variants identified to be under HLA-dependent selection in analysis Ib (86/98), the interaction effect between HLA and viral variants on VL was statistically insignificant, and they did not exhibit the anticipated virus-load pattern typically associated with escape mutations. This observation may be attributed to limitations in statistical power. However, it is also consistent with previous analyses indicating that moderate differences in viral fitness may often result in no or only weak variations in VL [13, 29].

Overall, we found a weak correlation between HLA-associated selection and predicted disruption of HLA binding ($\rho = 0.11$, $p = 0.38$). For 26/58 HLA-viral epitopes, the mutation was predicted to disrupt binding to the HLA allele. For the remaining pairs, e.g., Nef-Y135F: A*24:02, the HLA-dependent selection could not be explained by disruption of HLA binding. Thus, experimental characterizations are necessary to reveal the nature of these escapes, which may operate at different levels such as antigen processing or disruption of T cell receptor recognition.

We identified predominantly HLA/HIV-variant associations within the proteins Gag, Pol, and Nef (consistent with [12]). Despite being rather short proteins, Nef and Vpr exhibited numerous time-dependent associations. Relative to the number of possible combinations, we found for Vpr and Nef even more significant HLA/HIV-variant pairs than for Gag and Pol. Although Nef accumulates many mutations during the course of an HIV infection and these mutations hamper Nef's function in HLA downregulation [30], many of these mutations exhibit a different pattern than typical HLA escape mutations (as seen in Fig 2D). Similar deviations from this "escape pattern" were also observed for a high proportion of variants in Vpr, Rev, and Tat. These mutations appear to have had no or moderate costs on HIV replication, consistent with prior findings in Nef [31], while Gag or Pol were less tolerant to mutations.

Consequently, escape mutations in Gag/Pol are subject to rapid reversion in HLA-unmatched individuals [16]. Alternatively, variants like Rev57G or Nef85L may have been selected in the absence of HLAs and were initially stable escape variants that survived transmission and gradually accumulated in the population until becoming the consensus [24].

Even though we had good sequence coverage of Env in our study, the number of HLA-HIV associations was relatively low compared to the number of possible combinations. This can be attributed to several factors. For one, Env is highly variable, with three times more possible HLA/HIV-variant combinations than for example Pol, thus it may have fewer stable and conserved epitopes that can be presented to HLA molecules. Furthermore, much of the selection pressure on Env comes from antibody escape, so the lack of detection of HLA class I-dependent effects could be due to the fact that HLA class I-dependent selection plays a weaker role compared to other HLA class II or HLA-independent mechanisms [32,33]. This may result both in a lower number of HLA-HIV interaction pairs and a lower statistical power to detect them.

Prior research often excluded HLA class II alleles [12,13]. Our observations confirm that HLA class II alleles exert comparatively less immune pressure on HIV, with only six such alleles identified through our time-dependent screen. This supports that CD4$^+$ T cell responses may exert less direct influence on viral variants compared to CD8$^+$ T cell responses in the context of HIV infections. Consistent with Gabrielaite et al. [12], none of the HIV variants kept significant associations with the interaction between APD and HLA class II when adjusted for linkage disequilibrium between HLA alleles affecting the same viral variants.

A comparison of our findings with those of previous studies and the Los Alamos Immunology database revealed that 32 of the 98 pairs identified through our time-dependent analysis were to our knowledge not previously described (S2 Table) and are not included in [12,13,34,35]. This may be attributed to the fact that our systematic screen had a high sequence coverage across the whole HIV genome, even short genes like *vif*, *vpr*, *tat*, and *rev*. Additionally, we included HLA class II alleles in our analysis. However, it is important to note that these associations should be interpreted with caution, as they may be the result of linkage disequilibrium, as mentioned in the previous paragraph.

Our study bears several limitations. Firstly, although the SHCS comprises individuals from diverse ethnic backgrounds, the majority of participants are of European origin, White ethnicity, and HIV-1 subtype B sequences. Consequently, the generalizability of our results may be limited, as allele frequencies and HIV-1 subtypes can vary among populations [36,37]. Furthermore, the analysis of non-B subtype samples was hindered by small sample sizes, potentially causing us to miss true effects in this population. Nevertheless, all analyses were adjusted for population structures to reduce spurious associations caused by underlying demographic parameters. Secondly, we included plasma and proviral cellular-derived NGS sequences, introducing potential biases, but we show that excluding the proviral sequences did not make a difference. Thirdly, we solely analyzed consensus HIV sequences, disregarding within-host diversity. For future studies, especially those involving longitudinal analyses, conducting investigations of intra-host evolution could offer valuable insights.

In conclusion, we have developed a novel viral diversity-based approach to systematically identify HLA-dependent selection across the HIV genome and map potential escape mutations, and we validate these mutations using longitudinal data and binding predictions. Our approach provides an effective assessment of viral adaptation to their host's genome in the course of infection. As it is based on cross-sectional data, it allows screening for such adaptations in large populations and genome-wide screens. The approaches used here are likely to be applicable to other chronic infections in which host factors exert similar selection pressures driving the emergence of escape mutations and viral adaptation.

## Methods

### Ethics statement

We included participants enrolled in the Zurich Primary HIV Infection Study, an open-label, non-randomized, observational, monocenter study at the University Hospital Zurich, Switzerland (www.clinicaltrials.gov, ID NCT00537966; approved by the ethical committee (KEK-ZH-Nr. EK-1452) [38–40], and the Swiss HIV Cohort Study (SHCS), an ongoing, multi-center, prospective, longitudinal, observational cohort of people with HIV in Switzerland [21]. The SHCS was approved by the participating centers' local ethical committees: Ethikkommission beider Basel ("Die Ethikkommission beider Basel hat die Dokumente zur Studie zustimmend zur Kenntnis genommen und genehmigt."); Kantonale Ethikkommission Bern (21/88); Comité départemental d'éthique des spécialités médicales et de médecine communautaire et de premier recours, Hôpitaux Universitaires de Genève (01–142); Commission cantonale d'éthique de la recherche sur l'être humain, Canton de Vaud (131/01); Comitato etico cantonale, Repubblica e Cantone Ticino (CE 813); Ethikkommission des Kantons St. Gallen (EKSG 12/003); Kantonale Ethikkommission Zürich (KEK-ZH-NR: EK-793). All participants signed written informed consent [21]. We included participants with genotyped HLA alleles, at least one next-generation HIV sequence (NGS), and ART-naïve virus load. For the statistical analyses I-III, the number of participants and samples were further restricted (see below; Fig 1A).

### Human genotyping

Human leukocyte antigen (HLA) alleles and the first ten Principal Components (PC) were derived from human-genome single-nucleotide-polymorphism data for 6,607 SHCS participants genotyped in the setting of previous SHCS genetic studies. Genotyping data was processed and imputed separately for each batch as described previously [41], then merged with PC calculated using PLINK 2.00a5LM [42,43]. Classical HLA alleles at the 4-digit level were imputed separately per genotyping batch using SNP2HLA with the Type 1 Diabetes Genetics Consortium reference panel for samples of European ancestry [44] or HLA-TAPAS for all non-European ancestry participants [45]. For participants genotyped more than once, we retained the sample with the least amount of missing common variants.

### HIV sequencing, alignment, and variant calling

HIV-genome sequences were obtained by Illumina MiSeq next-generation sequencing (NGS) (see [46–50] for details) from plasma RNA and proviral DNA (see sensitivity analysis excluding the latter). Sequences were assembled *de-novo* with SmaltAlign [51]. The nucleotide bases were called for positions with a reading depth ≥20. Consensus sequences were generated and retained if ≥20% of a gene was covered. For statistical analyses I and II we selected the participant's first sequence. If several sequences were available for this date, we prioritized sequences from plasma viruses and then sequences with the highest genome completeness. Amino acid sequences were generated for each HIV gene using BLAST 2.13.0 [52] to select the respective genes on nucleotide level using a reference panel of 459 HIV sequences from the Los Alamos HIV Sequence Database [53], followed by translation with MACSE 2.05 [54]. For the translation and the multiple sequence alignment of these amino-acid sequences, the HIV-1 HXB2 reference genome (GenBank accession number K03455.1) was used. The nomenclature for HIV amino acid variants presented in this study follows the HXB2 numbering system. The alignment was done separately for each analysis and protein using MAFFT 7.520 [55] and MUSCLE 3.8.31 [56].

For statistical analysis III, participants were included if at least two treatment-naïve (grace period 7 days) sequences were available at different time points at least 30 days apart. Statistical analyses I-III were restricted to HIV-1 subtype B (determined by COMET [57] and REGA [58] from whole-genome nucleotide sequences). HIV-1 non-B subtypes were included in sensitivity analyses.

## Average pairwise diversity

To approximate the time since infection, we used the average pairwise viral diversity (APD) calculated from the third codon positions of the *pol* and *gag* genes, as this has been shown to be the most accurate [22]. For variants located in the *gag* gene, we used a *gag*-based APD, while for other variants we used a *pol*-based APD. APD values greater than 0.05 were excluded, as they are typically linked to superinfection [22] and thus unsuitable for assessing infection time. Since the use of APD as a proxy for infection time has been derived and validated solely for ART-naïve sequences, all assessments utilizing APD are limited to sequences obtained prior to ART initiation (participants in statistical analysis Ib, see Fig 1).

## HIV viral load

Chronic-phase HIV-1 viral load (VL) was calculated based on HIV-1 RNA measurements sampled during untreated chronic infection; i.e., the period from 30 days [59] after HIV diagnosis until 30 days before the first AIDS symptoms [60] or until ART initiation (maximally seven days after). For a given viral sequence, the corresponding VL was calculated as the mean $\log_{10}$-transformed HIV-1 RNA copies/ml plasma over ±180 days around its sampling date.

## Statistical analyses

To assess the interplay between HLA, viral amino-acid variants, and VL, we performed three series of analyses (Fig 1C): For analysis I, we systematically screened for HLA-dependent selection of viral variants by multivariable logistic regression models with the presence of HIV variant as the outcome and HLA-APD interaction terms (Fig 1B, Ib). We first reduced eligible HLA/HIV-variant combinations to those for which the frequency of HLA and HIV variants yielded a statistical power ≥80% for detecting an HLA/HIV-variant association with an odds ratio (OR) of three. We then tested the association between HLA and HIV variant (by Fisher's exact test), adjusted for multiple testing by Benjamini-Hochberg, retaining those pairs with a false discovery rate (FDR)<0.2 [61]. We chose a liberal FDR to avoid rejecting true pairs, as the selected pairs were further validated. In these pairs, we then used logistic regression models to test the association between the presence of the viral variant (1 = present, 0 = absent), and the HLA-allele (Ia), APD, HLA-APD interaction (Ib), and the first ten human-genome PCs. APD was only added to the models in analysis Ib and not Ia, as the participants needed to be ART-naïve. HLA/HIV-variant pairs with significant HLA-APD interactions (FDR-corrected p<0.05) were considered in analyses II and III. Analysis II evaluated the impact of HLA/HIV-variant interaction on VL (Fig 1B, II) using multivariable linear regressions adjusted for APD and the first ten human PCs. Analysis III assessed if participants with particular HLA alleles acquire the corresponding viral mutations at an increased rate, using longitudinal ART-naïve HIV sequences in Cox-proportional-hazards models (Fig 1B, III). In all statistical analyses I-III, the viral variant was compared to the most prevalent variant at a given position. For instance, a variant corresponding to residue X was compared to all non-X variants at a given position. Statistical analyses were conducted in PLINK 2.0 [42,43], and R 4.3.1 [62].

Several sensitivity analyses were performed. We tested whether proviral sequences biased our results, by repeating analysis Ib with proviral sequences previously filtered out. We tested

the effect of the HIV-1 subtype, by using all ART-naïve non-B sequences to reexamine the HLA-APD interaction for the pairs identified in the analysis Ib (additionally adjusting for the first ten viral PCs generated across the whole viral genome to exclude subtype-dependent effects). We assessed the effect of linkage disequilibrium between HLA alleles, by testing HLA alleles associated with the same variant in multivariable analyses.

## Computational epitope prediction

We used epitope prediction implemented in NetMHCpan-4.1 to assess the effect of HIV variants on class I HLA binding for the HLA/HIV-variant pairs identified in analysis Ib (Fig 1B). NetMHCpan-4.1 predicts the binding of peptides (9-mer) to MHC class I molecules using neural networks [63]. For a given HLA/HIV-variant pair, we assessed the predicted mass-spectrometry eluted ligands (EL) rank differences between the consensus and the mutated peptide (for all 9-mers containing the viral variant). Viral peptides with a binding rank below 2% or 0.5% were classified as binders or strong binders, as previously described [63]. Only epitopes with at least one binding rank below 2% were included and reported.

## Supporting information

**S1 Fig. Analysis plan with the number of pairs with significant results included. A)** HLA/HIV-variant pairs were selected if power>0.8 and Fisher's exact test p-value<0.2 (FDR-corrected). Statistical analyses performed: [Ia] Presence of HIV variants as function of HLA alleles. [Ib] Presence of HIV variants as function of HLA alleles, APD, and interaction between HLA and APD. [II] VL levels as function of HLA/HIV-variant pairs (identified as significantly associated in [Ib]); and [III] longitudinal survival analysis of HLA/HIV-variant pairs identified in [Ib]. Only samples from ART-naïve participants were used for the grey-shaded analyses ([Ib], [II] and [III]). **B)** Same structure as panel A, but with the number of pairs indicated that show significance in the statistical test or have significant associations in the analyses. The format is x/y, where x stands for the number of pairs with significant associations (or [†] interaction terms) and y stands for the number of pairs that were included in that particular analysis. For analysis III, 40 pairs were excluded from the analysis due to low sample size and lack of events, explaining the discrepancies between the y number in II (98) and III (58).
(TIFF)

**S2 Fig. Pairs with significant associations between HIV amino acid variants and HLA alleles.** 433 pairs grouped by HIV genes and arranged by odds ratios (ORs). ORs and 95% confidence intervals derived from Fisher's exact test. Color coding is based on HLA genes, with class I represented in gray and class II represented in red.
(TIFF)

**S3 Fig. Human leukocyte antigen-wide association analysis on HIV viral load.** Associations between human leukocyte antigen (HLA) alleles and viral load are calculated in multivariate linear regressions and are grouped by HLA genes. P-values are shown as -$\log_{10}$ transformation, sample size (n) = 3,676. Two thresholds are depicted: Bonferroni threshold ($p = 0.05/n$; dotted line) and $p = 0.05$ (dashed line).
(TIFF)

**S4 Fig. NetMHCpan-4.1 binding predictions: mutation vs. consensus of HIV variant.** The coloration of odds ratios (OR) is derived from the HLA-APD interaction model (analysis Ib). Pairs with no change in binding rank (strong, weak, no binding) by NetMHCpan-4.1, have lighter shading.
(TIFF)

**S5 Fig. Correlation analyses of the interaction effects between HLA and APD on HIV vs. A)** interaction effects of HLA/HIV-variant pairs on VL (analysis II estimates), **B)** Hazard ratios of occurrence of viral variant in presence/absence of HLA (analysis III cox-proportions test estimates), **C)** EL rank ratios (EL rank mutation/ EL rank consensus; binding predictions derived from NetMHCpan), **D)** Proviral sequences (n = 61) excluded, estimates of the interaction effects between HLA and APD on HIV, **E)** Non-B subtypes (A, C, AE, and others) estimates of the interaction effects between HLA and APD on HIV. **F)** All subtypes (A, B, C, AE, and others) estimates of the interaction effects between HLA and APD on HIV. Dark versus light shading indicates significance/ change on y-axis variables. Spearman's rank correlation $\rho$, $p$-value, and linear regression lines with 95% confidence intervals were calculated either for each subgroup alone (panel A, C) or for all together (panel B, D, E, F).
(TIFF)

**S6 Fig. Linkage disequilibrium assessment among HLA alleles that are associated with the same HIV variant.** Two multivariable logistic regression models were compared—one from analysis Ib (triangle form) and the other that includes additional interactions with other HLA alleles with the same viral variant (round form). Error bars reflect confidence intervals for the second model with linkage. Color coding is based on HLA genes, with class I represented in gray and class II represented in red.
(TIFF)

**S1 Table. An overview of sequence coverage and number of combinations HLA and HIV variants per gene.** The sequence coverage shows the number of participants/sequences available per gene, whereas all other columns refer to the different filtering steps. See Methods for more details on procedure.
(DOCX)

**S2 Table. All 98 pairs with significant interaction effects between HLA and APD on the HIV variant.** Odds ratios (OR) for the interaction effect of HLA and APD on the viral variant, estimates (β) of the interaction effect between HLA allele and HIV variant on viral load (VL), Hazard Ratios (HR) of the occurrence of the viral variant in presence/absence of HLA, NetMHCpan EL rank binding predictions before and upon mutation in different positions (pos), and its respective HIV 9-mer consensus epitope. 35 HLA-HIV associations pairs were newly described, all others are listed in the Publication reference list and Los Alamos (LA). All ORs and estimates having $p$-values<0.05 or changing binding categories are marked in bold.
(DOCX)

**S3 Table. Characteristics of SHCS participants included.** The number of participants used for cross-sectional (I and II) and longitudinal analysis (III) is shown for each analysis, with further information on the percentage of subtype B and non-B and the percentage of ART-naïve participants in the study population.
(DOCX)

**S1 Acknowledgements. Membership of Swiss HIV Cohort Study (SHCS).**
(DOCX)

## Acknowledgments

We thank the patients for participating in the Swiss HIV Cohort Study (SHCS), the study nurses, physicians, data managers, and administrative assistants. We thank the BEEHIVE

collaboration for providing the sequence data of the SHCS participants included in the BEE-HIVE study.

## Author Contributions

**Conceptualization:** Nadia Neuner-Jehle, Marius Zeeb, Huldrych F. Günthard, Chloé Pasin, Roger D. Kouyos.

**Formal analysis:** Nadia Neuner-Jehle, Marius Zeeb, Christian W. Thorball, Paul Frischknecht, Chloé Pasin, Roger D. Kouyos.

**Funding acquisition:** Nadia Neuner-Jehle, Chloé Pasin, Roger D. Kouyos.

**Resources:** Christian W. Thorball, Jacques Fellay, Karin J. Metzner, Paul Frischknecht, Kathrin Neumann, Christine Leeman, Andri Rauch, Marcel Stöckle, Michael Huber, Matthieu Perreau, Enos Bernasconi, Julia Notter, Matthias Hoffmann, Karoline Leuzinger.

**Supervision:** Chloé Pasin, Roger D. Kouyos.

**Writing – original draft:** Nadia Neuner-Jehle, Marius Zeeb, Huldrych F. Günthard, Chloé Pasin, Roger D. Kouyos.

**Writing – review & editing:** Nadia Neuner-Jehle, Marius Zeeb, Christian W. Thorball, Jacques Fellay, Karin J. Metzner, Paul Frischknecht, Kathrin Neumann, Christine Leeman, Andri Rauch, Marcel Stöckle, Michael Huber, Matthieu Perreau, Enos Bernasconi, Julia Notter, Matthias Hoffmann, Karoline Leuzinger, Huldrych F. Günthard, Chloé Pasin, Roger D. Kouyos.

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
