## [Decision Letter · Decision Letter 0]

20 Mar 2024

Dear Dr. Kouyos,

Thank you very much for submitting your manuscript "Using Viral Diversity to Identify HIV-1 Variants Under HLA-Dependent Selection in a Systematic Viral Genome-Wide Screen" for consideration at PLOS Pathogens. As with all papers reviewed by the journal, your manuscript was reviewed by members of the editorial board and by several independent reviewers. In light of the reviews (below this email), we would like to invite the resubmission of a significantly-revised version that takes into account the reviewers' comments.

Each of the reviewers focused on a different issue in your paper. You should carefully consider each of these comments in a revision.

We cannot make any decision about publication until we have seen the revised manuscript and your response to the reviewers' comments. Your revised manuscript is also likely to be sent to reviewers for further evaluation.

Sincerely,

Ronald Swanstrom

Section Editor

PLOS Pathogens

Michael Malim

Editor-in-Chief

PLOS Pathogens

orcid.org/0000-0002-7699-2064

Each of the reviewers focused on a different issue in your paper. You should carefully consider each of these comments in a revision.

Reviewer's Responses to Questions

**Part I - Summary**

Reviewer #1: Neuner-Jehle and colleagues conducted a large scale screen of HIV sequence and HLA genotype data to identify HLA-mediated polymorphisms while accounting for the effect of time since infection. This is a novel way to address the question of HIV’s adaptation to HLA alleles that takes advantage of a large and well-described cohort of persons living with HIV-1 in Switzerland. This is an interesting study that provides a useful methodology for the field.

The figures are data-heavy but sufficiently clear. My main comment is about the interpretation of the findings. I think the results highlight a limited effect of HLA-dependent selection. While I understand why the authors may not have wanted to emphasize results perceived as negative, I think it would have been useful to discuss further what the small number of significant associations vs. the large number of potential associations mean.

Reviewer #2: This study took a novel approach to enable a comparison of a very large population of individuals enrolled in the Swiss HIV cohort study to assess the host-virus interactions, by a clever utilization of the average pairwise viral diversity (APD) as a proxy for time since infection. They were able to look for associations in amino acid mutations across the HIV proteome whole viral genome sequences and (HLA) genotype and viral RNA load (VL) during untreated chronic infections. They found 98 HLA/viral-variant pairs over time, 12 of which associated with an impact on VL. 48/98 were supported by computational HLA-epitope predictions.

This is an valuable approach for studying the impact of CD8 T cells, the statistical strategies are appropriate, and as they note in the discussion, “This approach stands out by harnessing the potential of cross-sectional data and hence bridging between the within-host and epidemiological perspectives.”

While paper seems elegant in terms of bioinformatics, the authors could improve the paper by providing more biological context from the literature regarding their most interesting associations.

Reviewer #3: This is an interesting analysis in which the authors assess potential time-dependent effects of HLA-associated HIV adaptation using cross-sectionally sampled near full length viral genomes.

The authors emphasize their development and use of a viral genetic diversity score (average pairwise diversity, "APD") as a proxy for time since infection. While this measure appears to be useful, as stated, because HIV genetic diversity does tend to increase over the course of untreated infections, I am bit puzzled by the authors' insistence that this approach and measurement is such a novel development. From the paper it seems (to this reviewer, at least) that the authors are focused on the APD itself, as the major new tool, rather than the fact that this tool is required only because the authors are attempting to use single timepoint samples from individuals without good data on time since infection in order to infer time since infection. And that once individual time since infection can be inferred, or APD used as a proxy, then longitudinal patterns of HIV evolution can be identified. I might even recommend a different title that highlights that approach, rather than the generally descriptive current title. Something along the lines of, "Using intra-host genetic diversity to approximate time since infection in studies of HIV and HLA interaction."

Including cross-sectional and longitudinal analyses in this paper does fill out the scientific story well.

Overall, with regards to findings, the results are broadly confirmatory, in that HLA escape mutations decline in the absence of the specific HLA and increase in the presence of the specific HLA, and B-5701 is, once again, shown to be the HLA with the strongest impact on HIV disease (VL) and viral evolution. Thus it is slightly a methods paper, rather than a findings paper, and this perhaps should be the emphasis.

**Part II – Major Issues: Key Experiments Required for Acceptance**

Reviewer #1: It appears that the statistical analysis is based on the Ia dataset, ie a dataset that has a third of participants who are on ART. While I understand that this cohort can be used to look for associations between HIV sequences and HLA alleles, it is problematic to use this dataset for the time dependent analysis. Because the effect of time is measured using a proxy metric that corresponds to diversity at the third codon position, it seems important that this measure be calculated on sequences isolated from participants with uncontrolled viremia. Unless all the participants on ART are failing therapy, I think the participants on ART should be excluded and the analysis to look for associations should be based on the Ib dataset of ART naïve individuals.

The results did not lead me to the same conclusions as the authors. I see limited effects with no association for 335 HIV/HLA pairs and only 98 associations, including just 12 with an effect on viral loads. In contrast, the abstract mentions ‘numerous novel interactions’ and ‘frequent HLA-mediated selection’. It would be useful give more precise numbers for what numerous corresponds to as the results highlighted only a few examples. I think the ‘textbook’ scenario of escape mutation followed by a viral load effect has been clearly demonstrated in only a few cases in the literature, and I felt that the data supported that this is indeed a rare scenario and that in most cases the interactions between sequence/HLA diversity, time and viral loads are complex and difficult to identify.

It would be useful to provide more details on the APD and specifically on how it relates to time since infection as previously described in Carlisle et al. and in the context of the current lengths of infections in the participants studied.

Reviewer #2: I think they should provide background to validate that the HLA associated mutations are in actual epitopes when possible, by comparing more directly to the literature. Though the predictions they have implemented are useful, more should be done. I give some examples in Part II for an easy way to approach this starting with the Los Alamos HIV database. In the process of doing this they may gain some insights into the proposed escape mutations they observe and the biological outcomes of responses to particular epitopes. They could do this systematically for all of Table S1, but in particular, it would be helpful to draw out more cleanly and detail what is known regarding the 12 examples that were associated with VL differences would be valuable. I think this would make the paper both more readable and interesting to the immunology community.

Reviewer #3: (No Response)

**Part III – Minor Issues: Editorial and Data Presentation Modifications**

Reviewer #1: Abstract line 12: should it be 28 of 58 instead of 28/98 as described in the results (given that 40 pairs were excluded for lack of events)

Line 66- Table 1- Figure 1: the numbers do not match between the table, figure and text. It would be useful to describe what the analysis numbers (eg Ia) correspond to the first time they are described or the corresponding numbers are described.

Line 91 + discussion. The discussion describes effects in the different genes and that associations in some genes appear to be over-represented but they do not discuss why associations in Env are rare compared to the size of Env. It would be useful to describe the results for Env in more details in the paragraph starting line 91. Could the authors also provide more details on the sequence coverage of each gene, i.e. did all participants have env sequences used in the analysis?

Lines 113-125: How were the 8 associations described in Figure 2B selected?

Line 155. Verify the IQR value (maybe 1.02 instead of 2.02].

Line 178-180: For the computational epitope predictions, is there a possibility to use a continuous measure (or fold change) to see if there is a difference associated with a mutation that might be smaller than what is seen with the categories.

Figure 2C: typo in ‘occurrence’ in the y axis label

Reviewer #2: 1. These two papers would be worth mentioning in the introduction. While the McMichael group did a genome wide analysis with longitudinal samples and very detailed experimental validation of the variants and CTL responses that were observed early in infection, such experiments are constrained by their complexity to small number of people, and the less detailed broader view is of course also of interest. They might consider exploring the relative entropy of the mutations they observe associated with HLA and viral load, building on what Liu et al. observed.

Vertical T cell immunodominance and epitope entropy determine HIV-1 escape. Liu MK et al. J Clin Invest 2013 Jan;123(1):380-93. doi: 10.1172/JCI65330. Epub 2012 Dec 10. PMID: 23221345

The first T cell response to transmitted/founder virus contributes to the control of acute viremia in HIV-1 infection. Goonetilleke N et al., J Exp Med. 2009 Jun 8;206(6):1253-72.

doi: 10.1084/jem.20090365. Epub 2009 Jun 1.

PMID: 19487423

2) Line 118: How do they interpret negative interaction terms? (line 118)

3) Line 127 and 130: Regression models, HLA-B*57:01 and B*57:03 were highly associated with lower VL. This was the strongest interaction, and needs to be put in a better literature context, as do the full set of 12 HLA/HIV-variant pairs in total with a significant interaction effect on VL.

Also, they say the 12 cases are illustrated in Fig. 3D and 3E, and perhaps I am missing something but I don’t see 12 highlighted in Fig. 3D and 3E, and I’m not getting how to link between these figures and the table S1, where 12 they are noted.

A think a figure that really specifically highlights the 12 cases would be helpful.

4) Line 166-181 and Table S1: this is a more detail relevant to the point also mentioned in Part II:

“We used computational MHC class I binding prediction, to assess whether the identified HLA-1

allele/viral-variants pairs could be explained mechanistically”.

It would be helpful to list the specific proposed epitopes that changed HLA binding based on epitope prediction to the table. They check all 9 mers spanning the site of interest, but what do they think is actually the likely epitope(s) based on the EL predictions? Could they write them out?

As the binding predictions are not really validation, rather just support for the epitopes overlapping the variable sites, further validation with known experimentally defined epitopes from the literature whenever possible would provide a clearer proof.

So I think Table S1 would benefit from more information added, or alternatively an additional table could be added.

A quick way to begin to explore this literature would through the immunology database at Los Alamos, where they could cross-check their observed HLA/mutational associations and predictions against both known epitopes and experimentally determined escape patterns.

An easy way to at least begin to look more deeply is to compare supplemental table S1 to the epitope maps at the Los Alamos database. I think they would find examples of mapped epitopes, escape mutations, and responses associated with clinical outcomes that might enrich the context of mutation listings in their paper.

The clickable known epitope CD8 T cell maps can be found here:

https://www.hiv.lanl.gov/content/immunology/maps/maps.html

Here is Gag:

https://www.hiv.lanl.gov/content/immunology/maps/ctl/Gag.html

Their very first Table S1 entry, Gag18R:A*30:01 has an interesting characterization in previous literature:

Gag18R:A*30:01

Clicking on the epitope map takes you to this link:

https://www.hiv.lanl.gov/mojo/immunology/search/ctl/results.html?protein_name=Gag&start=18&end=26&hla_id=467

But a better example is a little further down the table, Gag264K~B*27:05, is embedded in a very interesting very well studied well studied B*2705 epitope. Their Gag264K is known to be associated with partial escape.

https://www.hiv.lanl.gov/mojo/immunology/search/ctl/results.html?protein_name=Gag&start=263&end=272&hla_id=3161

5) Line 328. “For the translation and the multiple sequence alignment of these amino-acid sequences, the HIV-1 HXB2 reference genome (GenBank accession number K03455.1) was used. alignment was done separately for each analysis and protein using MAFFT 7.520(48) an MUSCLE 3.8.31(49)."

Is HXB2 also the numbering system you used to describe mutations?

6) Are the HIV sequences all submitted to Genbank? I didn't see the accession numbers, excuse me if this was noted and I missed it. A full alignment of all sequences to accompany the paper would also be helpful.

7) To name their mutations they provide a number with an amino acid, but could they add the amino acid before and after a change? Is Gag18R actually Gag K18R, and Gag264K actually Gag R264K for example? If so, that kind of label would be clearer.

Reviewer #3: (No Response)

PLOS authors have the option to publish the peer review history of their article (what does this mean?). If published, this will include your full peer review and any attached files.

Reviewer #1: **Yes: **Morgane Rolland

Reviewer #2: No

Reviewer #3: No
---

## [Decision Letter · Decision Letter 1]

27 May 2024

Dear Dr. Kouyos,

Thank you very much for submitting your manuscript "Using Viral Diversity to Identify HIV-1 Variants Under HLA-Dependent Selection in a Systematic Viral Genome-Wide Screen" for consideration at PLOS Pathogens. As with all papers reviewed by the journal, your manuscript was reviewed by members of the editorial board and by an independent reviewer. The reviewer appreciated the attention to an important topic. Based on the reviews, we are likely to accept this manuscript for publication, providing that you modify the manuscript according to the review recommendations.

The manuscript received a wide range of reviewer responses in the initial review and for that reason we asked one of the reviewers to consider your responses. While most of the changes were appropriate, the reviewer raised several important points that need to be addressed. Given that this is a very knowledgeable reviewer I believe close attention to these points will improve your manuscript.

Please prepare and submit your revised manuscript with responses within 30 days. If you anticipate any delay, please let us know the expected resubmission date by replying to this email.

Sincerely,

Ronald Swanstrom

Section Editor

PLOS Pathogens

Ronald Swanstrom

Section Editor

PLOS Pathogens

Michael Malim

Editor-in-Chief

PLOS Pathogens

orcid.org/0000-0002-7699-2064

Reviewer Comments (if any, and for reference):

Reviewer's Responses to Questions

**Part I - Summary**

Reviewer #1: The revised version of the manuscript is improved. However, when rereading it, there are still things that are confusing regarding which participants are included in which analyses. It is unclear why they report the number of non-B sequences in Table 1 if the analyses are only done on a subtype B dataset. Could this information be in a Supplementary Table?

It would be helpful to provide a definition of variant in their context. Based on the figure and on the description of ‘combinations of HLA alleles and HIV amino acid variants’, I had understood that they were identifying pairs as corresponding to an HLA allele and a residue X at specific position. However, the text on page 8 line 105 could suggest an alternate meaning: ‘Of the 433 remaining significant pairs, 329 showed a positive association, wherein the viral variant was more prevalent when the HLA allele was present (OR median [IQR]: 2.86 [2.19, 3.83], as shown in S2 Fig).’ This statement suggests that there is a reference to define a variant. Is it the consensus at each position that is the reference? Is it HXB2? Are they comparing the most common variant, e.g.. a variant that corresponds to the residue A to all the non-A variants for a given site+HLA?

Regarding the statement ‘Ancestral and population structure biases likely accounted for these associations’, did the authors perform any phylogenetic correction ? (see Bhattacharya and colleagues (Science, 2007; https://www.science.org/doi/10.1126/science.1131528) who showed that most of the associations in Moore and colleagues (Science, 2002; https://www.science.org/doi/10.1126/science.1069660) were spurious because they were due to founder effects). The authors should address the potential need for phylogenetic correction.

**Part II – Major Issues: Key Experiments Required for Acceptance**

Reviewer #1: (No Response)

**Part III – Minor Issues: Editorial and Data Presentation Modifications**

Reviewer #1: (No Response)

PLOS authors have the option to publish the peer review history of their article (what does this mean?). If published, this will include your full peer review and any attached files.

Reviewer #1: **Yes: **Morgane Rolland

Figure Files:

Data Requirements:

Reproducibility:

References:

---

## [Editor Report · Decision Letter 2]

2 Jul 2024

Dear Dr. Kouyos,

We are pleased to inform you that your manuscript 'Using Viral Diversity to Identify HIV-1 Variants Under HLA-Dependent Selection in a Systematic Viral Genome-Wide Screen' has been provisionally accepted for publication in PLOS Pathogens.

Best regards,

Ronald Swanstrom

Section Editor

PLOS Pathogens

Ronald Swanstrom

Section Editor

PLOS Pathogens

Michael Malim

Editor-in-Chief

PLOS Pathogens

orcid.org/0000-0002-7699-2064
---

## [Editor Report · Acceptance letter]

2 Aug 2024

Dear Dr. Kouyos,

We are delighted to inform you that your manuscript, "Using Viral Diversity to Identify HIV-1 Variants Under HLA-Dependent Selection in a Systematic Viral Genome-Wide Screen," has been formally accepted for publication in PLOS Pathogens.

Best regards,

Michael Malim

Editor-in-Chief

PLOS Pathogens

orcid.org/0000-0002-7699-2064